# LLaMA Rider: Spurring Large Language Models to Explore the Open World

## Abstract

Recently, various studies have leveraged Large Language Models (LLMs) to help decision-making and planning in environments, and try to align the LLMs' knowledge with the world conditions. Nonetheless, the capacity of LLMs to continuously acquire environmental knowledge and adapt in an open world remains uncertain. In this paper, we propose an approach to spur LLMs to explore the open world, gather experiences, and learn to improve their task-solving capabilities. In this approach, a multi-round feedback-revision mechanism is utilized to encourage LLMs to actively select appropriate revision actions guided by feedback information from the environment. This facilitates exploration and enhances the model's performance. Besides, we integrate sub-task relabeling to assist LLMs in maintaining consistency in sub-task planning and help the model learn the combinatorial nature between tasks, enabling it to complete a wider range of tasks through training based on the acquired exploration experiences. By evaluation in Minecraft, an open-ended sandbox world, we demonstrate that our approach **LLaMA-Rider** enhances the efficiency of the LLM in exploring the environment, and effectively improves the LLM's ability to accomplish more tasks through fine-tuning with merely 1.3k instances of collected data, showing minimal training costs compared to the baseline using reinforcement learning.

## 1 Introduction

Recently, significant advancements and successes have been achieved in the performance of Large Language Models (LLMs) in attaining human-like intelligence (OpenAI, 2023). Given the powerful capability of LLMs, many research works have started utilizing their abilities to assist intelligent agents in decision-making in the environments (Yao et al., 2023; Huang et al., 2022a; Li et al., 2022; Singh et al., 2023), and have found that LLMs possess a certain level of abilities for planning and accomplishing various tasks (Wang et al., 2023b). However, the knowledge that LLMs rely on comes from the language corpus used during pre-training, and there may be discrepancies between this knowledge and specific environments (Ahn et al., 2022).

To ground LLMs to environments, some studies design specific mechanisms through prompt engineering to provide information from environments for LLMs (Wang et al., 2023c; Yao et al., 2023; Wu et al., 2023; Zhu et al.,

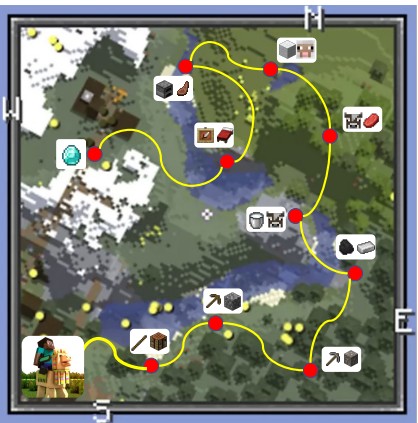

**Figure 1.** Spurring LLaMA to explore the open world.

2023; Liu et al., 2022). However, LLMs do not improve or acquire new knowledge in environments. Additionally, for more complex tasks, more complicated mechanisms and prompts are required, which results in high costs of LLM generation and reliance on strong models like GPT-4 (OpenAI, 2023) with enough knowledge (Wang et al., 2023a). Some other studies ground LLMs with finetuning (Yao et al., 2022; Deng et al., 2023; Xiang et al., 2023), but they usually require task-dependent datasets. Reinforcement Learning (RL) methods are also studied in the literature (Carta et al., 2023),

but these methods train LLMs as task-specific policies, and we found that RL methods are difficult to scale up to larger models or more complex tasks (see Section 5.2.2).

In this paper, we aim to enhance LLMs through their exploration in open-ended environments (Figure 1), like humans can adapt to new situations through practice. Previous studies have tried to update LLMs in embodied environments like BabyAI (Chevalier-Boisvert et al., 2019) and VirtualHome (Puig et al., 2018), but these world sizes are rather limited. Whether LLMs can improve their knowledge in more complicated open-ended worlds like Minecraft is still unknown (Fan et al., 2022; Guss et al., 2019). We think there are two major challenges here. First, in an environment like Minecraft, tasks are often complex and may involve many sub-tasks. At the same time, these long-horizon tasks often require each step to be carried out precisely, and a single error in the middle sometimes can negate previous progress. Besides, due to the high level of freedom, the action space can be large, while many actions may be invalid in different states. These reasons make it hard to collect successful task trajectories in the environment using random exploration as in previous works (Xiang et al., 2023; Li et al., 2022). The second challenge is that there can be a significant amount of tasks in such an open world, so training policies for specific tasks are not applicable in these environments. We hope that LLMs have the ability to perform multiple tasks and generalize to new tasks.

In response to these challenges, we propose **LLaMA-Rider** 🏇, a two-stage learning framework consisting of an exploration stage and a learning stage (Figure 2). We investigate how to spur LLMs to explore the environment themselves and collect successful experiences for learning. Compared to random exploration or search methods that can hardly work in complex environments, allowing LLMs to explore on their own in the environment can harness the inherent capabilities of the models, thereby enabling more effective discovery of successful experiences. We propose a multi-round feedback mechanism, which allows the LLM to revise its decisions by providing information about failed actions in the environment. This feedback-revision exploration mechanism is more efficient due to the capability of LLMs, as the draft decisions made are often related to task completion at first, and LLMs can effectively understand feedback information. Additionally, we use sub-task relabeling to help LLMs maintain consistency in sub-task planning.

In the learning stage, we process the collected experiences into datasets and use supervised finetuning (SFT) to train the LLM. In addition to the experience gained from successful tasks, we also collect experiences from partially completed sub-tasks, as some tasks are too difficult to accomplish in the environment in the exploration stage. Numerous tasks in open-ended environments often have compositionality, which means experiences from past tasks can frequently assist in completing other tasks. We propose to use sub-task relabeling of the collected experiences to improve data utilization while helping LLMs learn the compositionality between tasks.

We evaluate our method in MineDojo (Fan et al., 2022), a simulation platform for Minecraft. We use the basic skills trained by Plan4MC (Yuan et al., 2023) as the action space since the skills possess more semantics compared with primitive actions and are better aligned with LLMs. We use LLaMA-2-70B-chat (Touvron et al., 2023) in our experiments. Our experiments show that **LLaMA-Rider** can explore the environment efficiently with our feedback-revision mechanism, and can learn to complete tasks more effectively by finetuning on a collected dataset of only 1.3k in size, demonstrating much higher sample efficiency compared to RL methods. We also show the generalization ability of **LLaMA-Rider** in novel hard tasks.

## 2 RELATED WORK

### 2.1 LLM-BASED AGENTS

There is a large body of recent studies on LLM-based agents, which have delved into the capacities of LLMs for decision-making and are well summarized in the survey papers (Wang et al., 2023b; Xi et al., 2023). There are basically three ways to integrate LLMs into decision-making problems. First, using the code generation capabilities of LLMs, LLMs take in information from the environment and produce code that can interact directly within the environment (Liang et al., 2023; Singh et al., 2023). The second way is to employ LLMs for planning, following a concept similar to hierarchical RL (Ahn et al., 2022; Huang et al., 2022b; Wang et al., 2023c; Dasgupta et al., 2023). The third approach involves continually prompting LLMs or introducing memory modules to generate outputs

that can execute better strategies directly within a textual environment (Wei et al., 2022; Yao et al., 2023; Kim et al., 2023).

Minecraft, as a popular and challenging open-world benchmark, has also attracted substantial attention for the studies of LLM-based agents. DEPS (Wang et al., 2023c) introduces the descriptor, explainer, and selector for plan generation with the help of LLM. Plan4MC (Yuan et al., 2023) constructs a skill graph with the help of LLM and proposes a skill search algorithm for planning over the basic skills pretrained by reinforcement learning (RL). Moreover, to build LLM-based agents in Minecraft, Voyager (Wang et al., 2023a) leverages the code generation of LLMs, while GITM (Zhu et al., 2023) integrates LLMs with texted-based knowledge and memory.

However, in the aforementioned studies, LLMs do not update themselves from their interactions with the environment, so they can neither learn from nor adapt to the environment. Consequently, their potential applicability in specific environments is limited, as they can solely depend on the knowledge and capabilities gained during pre-training.

## 2.2 FINETUNING LANGUAGE MODELS IN ENVIRONMENTS

There are studies that ground Language Models (LMs) to environments with finetuning. PIGLeT (Zellers et al., 2021) integrates a neural symbolic dynamics model with an LM to learn natural language meaning grounded in physical interactions. Also focusing on the decision-making of LMs in embodied environments, LID (Li et al., 2022) uses expert trajectories to finetune a model that concatenates an LM with action decoders. They also propose active data gathering to collect experiences that mix random actions and policy-generated actions for exploration. Similarly, E2WM (Xiang et al., 2023) uses supervised learning to finetune LMs with the data collected by Monte Carlo Tree Search and random exploration. Additionally, GLAM (Carta et al., 2023) ground LMs in environments with online RL, but they train the LM into a task-specific policy, and the RL method suffers from low sample efficiency and high cost of training. Our work is different from existing work in that we spur the LLM *itself* to explore with feedback from the environment, and we target multi-task and generalization abilities in the open world.

## 3 PRELIMINARIES

### 3.1 LARGE LANGUAGE MODELS

LMs, which predict the probability of the $i$th token given inputs and the previously generated tokens $P_i = P(s_i|inputs, s_1, s_2, \cdots, s_{i-1})$, are used to generate a series of tokens by sampling from the probability of the token sequences $P(x) = \Pi_{i=1}^n P_i$, where $x$ can be considered as a random variable representing $n$ tokens in the token library. LLMs often have billions of weights and are trained from billions of tokens to enable them to achieve remarkable performance on generative tasks.

To finetune LLMs with full parameters requires remarkable compute resources. Fortunately, some techniques can help with efficient finetuning. Low-Rank Adaptation (LoRA) (Hu et al., 2022) involves the process of keeping the pretrained model weights fixed while introducing trainable rank decomposition matrices into every layer of LLMs. Original pretrained weights $W_0 \in \mathbb{R}^{d \times k}$ are augmented to $W_0 + \Delta W = W_0 + BA$, where $B \in \mathbb{R}^{d \times r}$ and $A \in \mathbb{R}^{r \times k}$. The matrices $A$ and $B$ are both trainable, with $A$ initialized to a normal distribution and $B$ initialized to zero. Moreover, QLoRA (Dettmers et al., 2023) adds quantization and paged optimizers to further reduce training compute costs. Quantization aims to transform input from a high-information representation into a low-information representation, such as converting FP32 to int8 to reduce memory usage.

### 3.2 PROBLEM STATEMENT

We consider an environment that can be formalized as a Partially Observable Markov Decision Process (POMDP) defined by tuple $\mathcal{M} = (\mathcal{S}, \mathcal{O}, \mathcal{A}, \mathcal{T}, \mathcal{R}, \gamma)$, where $\mathcal{S}$ is the environment state, $\mathcal{A}$ is the action space, $\mathcal{O}$ is the observation space, $\mathcal{T}$ is the transition function, $\mathcal{R}$ is the reward function, and $\gamma$ is the discount factor. Since we use LLMs as embodied agents, we assume a language vocabulary $\mathcal{V}$ and we can encode the observations and actions from the environment into natural language. Besides, we assume a goal space $\mathcal{G}$ and we can sample a task $\tau = (g, K), g \in \mathcal{G}$, where $g$

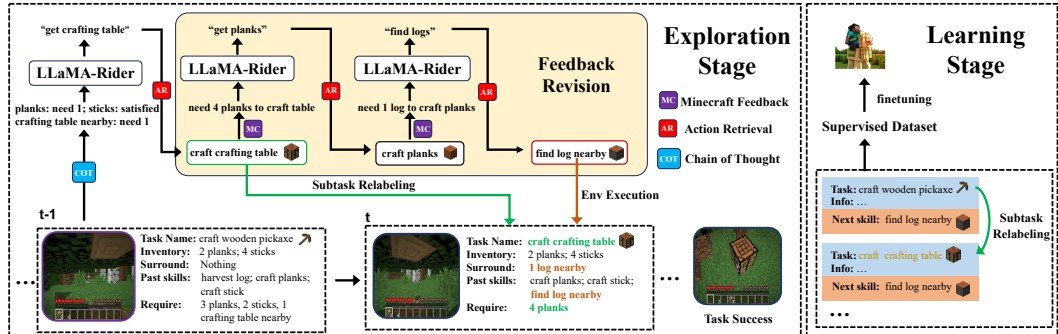

**Figure 2.** Overview of **LLaMA-Rider** . The framework consists of two stages. In the exploration stage, the LLM explores to accomplish tasks with the help of the feedback-revision mechanism and subtask relabeling. In the learning stage, the collected trajectories are formatted into a supervised dataset to finetune the LLM.

is the goal of the task and $K$ is the task information including task-relevant knowledge. We can also encode the task $\tau$ into task description $\tau^{text} \in \mathcal{V}^N$.

In this study, we explore the Minecraft simulator provided by MineDojo (Fan et al., 2022), which is an open-ended sandbox world. There is rich information in the observation space, but a big portion of it cannot be comprehended by LLMs such as game visuals. We extract the items in the agent's *inventory* and *field of view*, along with their quantities, and encode them into natural language sentences as the observations for LLMs: $o^{text} = (inv, fov) \in \mathcal{V}^N$. Primitive actions in the environment (e.g., move forward, turn right, click) have insufficient semantics which hampers the planning capability of LLMs. We use skill descriptions as the action space of the LLM agent noted with $a^{text} \in \mathcal{V}^N$.

### 3.3 Skills and Tasks in Plan4MC

We use the basic skills and tasks in Plan4MC (Yuan et al., 2023) in our experiments in MineDojo, since the basic skills have more semantic meaning than primitive actions. Plan4MC uses RL to train three types of basic skills: finding-skills, manipulation-skills, and crafting-skills. They then define 40 difficult tasks that can be completed with the trained skills. We define the action space of the LLM agent $\mathcal{A}_{text}$ as the descriptions of these basic skills.

## 4 Methodology

Our method is illustrated in Figure 2, which is a two-stage framework. We introduce the exploration stage and the learning stage respectively in the following.

### 4.1 Exploration with Feedback

**Prompt mechanism.** Unlike previous studies such as Voyager (Wang et al., 2023a) and GITM (Zhu et al., 2023) which use complex prompts to tweak LLMs to accomplish various tasks in open-ended worlds like Minecraft, our approach employs a straightforward prompt that makes LLMs provide the next action given input information about observation and task. This brings two advantages. First, it makes finetuning LLMs to learn from past experiences easy, considering the context-length limit of LLMs. Second, it reduces the cost of LLM generation.

Formally, the LLM serves as the policy $\pi(a_t^{text}|o_t^{text}, \tau^{text}, h_t)$. We provide the textual observation $o^{text}$, the task description $\tau^{text}$ and the history information $h$ in the input prompt to feed the LLM at each time step $t$, and the output of the LLM is the chosen action $a^{text}$. We find that if there are too many tokens of history information in the prompt, it will affect the output of the LLM. Therefore, in our experiments, we set $h$ to be the last three actions performed $h_t = (a_{t-3}^{text}, a_{t-2}^{text}, a_{t-1}^{text})$.

---

**Algorithm 1.** Feedback-revision

---

**Require:** $o_t^{text}, \tau^{text}, h_t, \pi_{LLM}, E, T$
**Ensure:** $a_t^{text}$
1: $a_t^{text} \sim \pi_{LLM}(\cdot|o_t^{text}, \tau^{text}, h_t)$
2: $f_t = E(s_t, a_t)$
3: **for** $i = 0$ to T **do**
4:     **if** $f_t = 0$ **then**
5:         **return** $a_t^{text}$
6:     **end if**
7:     $f_t \rightarrow f_t^{text}$
8:     $a_t^{text} \sim \pi_{LLM}(\cdot|o_t^{text}, \tau^{text}, h_t, f_t^{text})$
9:     $f_t = E(s_t, a_t)$
10: **end for**
11: **if** $f_t = 0$ **then**
12:     **return** $a_t^{text}$
13: **end if**
14: **return** 0

---

**Feedback-revision mechanism.** LLMs possess rich knowledge of the real world, but there is often a gap between the knowledge of LLMs and the specific environment to which they are applied. For example, which actions can be performed in the environment? What are the prerequisites for each action before execution? What conditions need to be satisfied for the completion of different tasks in the environment? What are the names of various items in the environment? LLMs often lack understanding of these questions, leading to decision-making errors. Previous studies ground LLMs to environments by searching through the action space (Xiang et al., 2023) or mix policy with random actions (Li et al., 2022) to collect experiences, or train LLMs with reinforcement learning (Carta et al., 2023). But these methods can hardly scale up to worlds with long-horizon tasks. They all do not provide environmental knowledge to LLMs but make LLMs explore through trial and error. We propose to spur LLMs to explore the world *themselves* with their reasoning capabilities by feeding them environmental feedback information and letting LLMs revise their decisions. LLMs can access environmental knowledge during this process, and the method makes use of LLMs' inherent ability to enhance the efficiency of exploration.

Formally, after the LLM produces an action $a_t^{text} \sim \pi(\cdot|o_t^{text}, \tau^{text}, h_t)$, a feedback information is generated by the environment $f_t = E(s_t, a_t)$, where $E$ denotes the environment, $s_t$ denotes the state, and $a_t$ denotes the primitive actions corresponding to $a_t^{text}$. If $f_t \neq 0$, which means the action causes an error, the feedback is processed by a prompt into $f_t^{text}$ and fed back to the LLM together with the previous input information, and the LLM would make a revision to produce a new action $a_t^{text'} \sim \pi(\cdot|o_t^{text}, \tau^{text}, h_t, f_t^{text})$. Then a new feedback is generated $f_t = E(s_t, a_t')$. This feedback-revision procedure can be repeated until $f_t = 0$ or the maximum number of allowed revisions $T$ has been reached which means the exploration has failed and the episode ends. The formalized approach of the feedback-revision mechanism can be seen in Algorithm 1.

**Subtask relabeling.** Long-horizon tasks in an open world are often composed of many subtasks. Since our input prompt is brief, limited information is provided. So the LLM planner may forget what subtask it is currently working on and opt to start completing other subtasks, resulting in failure to consistently complete one subtask. To solve this problem, whenever the LLM's output skill is accomplishing a subtask $\tau_s$ of the task $\tau$, we replace the task information $\tau^{text}$ in the input prompt with $\tau_s^{text}$ and keep it until $\tau_s$ is completed. This subtask relabeling provides another important benefit: some subtasks may have been met in the collected experiences as a simpler task or as a subtask of another task, so this method helps LLMs make use of previously learned experiences to solve new tasks.

**Action retrieval.** To match the output of the LLM with the action space, there are two major ways: feed the action list to the LLM or retrieve the action list based on the output. We find that feeding a lengthy list of actions as input to the LLM would affect its output to generate more unreasonable actions unrelated to the current task. Therefore, we use action retrieval to select an action from the action space that is closest to the output of the LLM. Additionally, we find that querying with token embeddings could cause retrieval errors since the action description often consists of only a

few words, e.g., "craft wooden planks" may be matched to "craft wooden sword" instead of "craft planks". We propose to use noun matching before embedding matching to alleviate this problem. Details of action retrieval can be found in Appendix C.

**Chain-of-thought (CoT) prompting.** In our experiments in Minecraft, we find that the LLM often makes decision mistakes due to insensitivity to the relationships between numbers. To enhance the efficiency of exploration, we integrate in-context learning and chain-of-thought prompting (Wei et al., 2022) that make the LLM compare the item numbers in the inventory and the requirements before making decisions. The prompt can be seen in Appendix B.3, and we only use it in the exploration stage for Minecraft.

## 4.2 FINETUNING LLMS WITH EXPERIENCES

**Dataset construction.** We compile task experiences of all tasks collected by the LLM from the environment into a supervised dataset, with the input be the task information and the observation $\mathbf{x} = (o_t^{text}, \tau^{text}, h_t)$, and the label be the action $\mathbf{y} = a_t^{text}$. In addition to success trajectories, we also include partial trajectories where a subtask is completed, since some tasks are too hard to accomplish during exploration, and the subtask experience may help the LLM to accomplish the whole task more easily. Besides, subtask experiences may also help the LLM solve some other tasks due to the compositionality. To better make use of the subtask information and encourage combinatorial generalization, we also use subtask relabeling to construct the dataset. Namely, if the LLM is solving a subtask $\tau_s$ of task $\tau$ at time step $t$ in a trajectory, we add the data ($\mathbf{x} = (o_t^{text}, \tau_s^{text}, h_t), \mathbf{y} = a_t^{text}$) into the dataset.

**Training.** With the dataset including experiences of various tasks in the environment, we train the LLM with supervised finetuning (SFT). We use QLoRA (Dettmers et al., 2023) to reduce memory usage, and more details can be found in Appendix A.

## 5 EXPERIMENTS

### 5.1 EXPERIMENTAL SETUP

**MineDojo environment.** We evaluate our proposed method on Minecraft based on the MineDojo (Fan et al., 2022) simulator. We use 30 difficult tasks in Plan4MC (Yuan et al., 2023) including three types: 10 log-based 🪵 tasks, 10 cobblestone-based 🧊 tasks, and 10 mob-based 🐷 tasks. The minimum number of planning steps provided by Plan4MC required for these tasks ranges from 2 to 30, with an average minimum of 11.5 steps. More details about the tasks can be found in Appendix D. We use 55 basic skills trained by Plan4MC and convert them to skill descriptions in natural language as the action space of the LLM. Note that the skill policies do not guarantee success, and the success rates of all the skills are provided in Appendix D. For each task $\tau = (g, K)$, the goal $g$ is the target item of the task and the knowledge $K$ is the requirement to achieve target $g$ in MineDojo. The feedback information $f_t$ from the environment is the requirements that are not met to execute skill $a_t$ in MineDojo. The prompt template for the LLM's input and the feedback can be found in Appendix B.

We define the subtasks of a task $\tau$ as the tasks $\tau_s = (g_s, K_s)$ whose goal $g_s$ is one of the requirements to achieve task $\tau$. For example, the task "craft bowl" has two subtasks "craft planks" and "place crafting table nearby". Note that some subtasks are simple so are not among the 30 difficult tasks for evaluation.

**LLM agent.** We use LLaMA-2-70B-chat (Touvron et al., 2023) as our LLM agent, which was recently released and has strong question-answering and instruction-following abilities. These abilities are important for the LLM to actively explore in the environment, and conversely, our method can also effectively make good use of its strong abilities to do something beyond question answering, namely exploring new environments.

**Baselines.** We compare with three baselines in our experiments. The first is **ChatGPT planner** (Ouyang et al., 2022), the interactive LLM baseline in Plan4MC, which uses a carefully designed prompt mechanism to make ChatGPT (GPT-3.5) propose skill plans. This baseline also uses the LLM to choose skills trained in Plan4MC for accomplishing tasks in Minecraft. Since ChatGPT

**Table 1.** Success rates in all tasks. **LLaMA-Rider Exploration** is tested for 5 episodes in log-based 🟫 tasks and 10 episodes in other tasks. All other methods are tested for 30 episodes. Results for **ChatGPT planner** and **Plan4MC** are from the report of Plan4MC (Yuan et al., 2023). **LLaMA-Rider Base** is **LLaMA-Rider** before finetuning.

| Task | LLaMA-Rider Exploration | LLaMA-Rider Base | ChatGPT planner | RL | LLaMA-Rider (ours) | Plan4MC |
|---|---|---|---|---|---|---|
| | *0.90* | 0.23 | 0.30 | 0.00 | **0.43** | 0.30 |
| | *1.00* | 0.37 | 0.17 | 0.00 | **0.67** | 0.30 |
| | 0.80 | 0.73 | 0.07 | 0.00 | **0.97** | 0.47 |
| | 0.60 | 0.67 | 0.00 | 0.00 | **0.77** | 0.23 |
| | 0.60 | **0.57** | 0.03 | 0.00 | 0.57 | 0.37 |
| | 0.00 | **0.67** | 0.00 | 0.00 | 0.60 | 0.43 |
| | 0.80 | 0.0 | 0.20 | 0.00 | 0.37 | **0.53** |
| | 0.60 | **0.77** | 0.47 | 0.00 | 0.60 | 0.37 |
| | 0.80 | 0.07 | **0.63** | 0.00 | 0.10 | 0.47 |
| | 0.00 | 0.03 | **0.73** | 0.00 | 0.27 | 0.70 |
| | 0.40 | 0.00 | 0.00 | - | 0.17 | **0.37** |
| | 0.10 | 0.00 | 0.20 | - | **0.57** | 0.47 |
| | 0.10 | 0.00 | 0.03 | - | 0.40 | **0.53** |
| | 0.20 | 0.00 | 0.13 | - | 0.10 | **0.57** |
| | 0.00 | 0.00 | 0.00 | - | 0.00 | **0.37** |
| | 0.00 | **0.13** | 0.00 | - | 0.07 | 0.10 |
| | 0.00 | 0.00 | 0.00 | - | 0.03 | **0.17** |
| | 0.00 | 0.00 | **0.07** | - | 0.03 | **0.07** |
| | 0.00 | 0.00 | **0.13** | - | 0.00 | 0.10 |
| | 0.10 | 0.00 | 0.10 | - | 0.07 | **0.20** |
| | 0.70 | 0.60 | 0.57 | - | 0.60 | **0.83** |
| | 0.30 | 0.50 | **0.76** | - | 0.57 | 0.53 |
| | 0.00 | 0.10 | 0.00 | - | 0.03 | **0.17** |
| | 0.00 | 0.10 | 0.00 | - | 0.07 | **0.13** |
| | 0.30 | **0.50** | 0.37 | - | 0.43 | 0.37 |
| | 0.00 | 0.00 | 0.00 | - | 0.00 | **0.07** |
| | 0.00 | 0.03 | **0.43** | - | 0.03 | **0.43** |
| | 0.00 | 0.00 | 0.03 | - | 0.00 | **0.20** |
| | 0.00 | 0.00 | 0.30 | - | 0.03 | **0.33** |
| | 0.00 | 0.00 | 0.00 | - | 0.00 | **0.13** |

possesses more accurate knowledge about Minecraft than LLaMA-2-70B-chat, by comparing with this baseline, we show whether our exploration-learning framework can enable an LLM to adapt to a new environment and outperform a stronger language model. The second is **RL** where we use the training framework proposed in GLAM (Carta et al., 2023) and use their default language model T5 (Chung et al., 2022). We try our best to fit GLAM into the Minedojo environment but we have to constrain the action space to include only the necessary actions to reduce sample complexity. The detailed implementation is described in the Appendix E. The third is **Plan4MC**, where they construct a skill graph and use depth-first search (DFS) for planning over basic skills. This baseline ensures that the planning is correct. Thus, it can be seen as an upper bound of our method. However, we note that our method may outperform Plan4MC in some tasks. We speculate this is because Plan4MC does not always generate the optimal plan in terms of planning steps, though the plan is correct.

## 5.2 EVALUATION

We set the maximum number of revisions as $T = 5$ for which we find can best balance the efficiency and success rate of the LLM's exploration for all tasks. Since the log-based 🟫 tasks are easier, we only perform 5 episodes of exploration, where we make the **LLaMA-Rider** explore for 10 episodes for the rest 20 tasks, so that the experience collected from different tasks be in similar quantities.

For the task "craft stick ✏" and "place crafting table 🟫 nearby", we change the biome to forest in the exploration stage to improve the chance of finding logs 🟫. The results are shown in Table 1.

### 5.2.1 EXPLORATION OF LLAMA-RIDER IN MINECRAFT

**LLaMA-Rider Exploration** shows the LLM's ability to explore in Minecraft to accomplish different tasks with our designed prompt combined with the feedback-revision mechanism. Compared with **ChatGPT planner** which is based on a powerful LLM with more Minecraft knowledge (see Appendix F), **LLaMA-Rider Exploration** can obtain successful experiences more effectively without finetuning in log-based 🟫 tasks and has comparable performance in the other tasks. This can be attributed to our feedback-revision mechanism, which provides more environment information for the LLM to acquire knowledge alignment, and the CoT prompt that mitigates the LLM's numerical comparison issue. Besides, the success rates in stone-based 🟫 tasks and mob-based 🐷 tasks demonstrate that it is difficult for LLMs to solve long-horizon complex tasks in environments just rely on prompt engineering, reflecting the importance for LLMs to update with environmental experiences to adapt.

### 5.2.2 ENHANCING LLM WITH ENVIRONMENTAL EXPERIENCES

**Performance in explored tasks.** We collect trajectories that the LLM achieves success in the whole tasks or subtasks and process them into a supervised dataset of 1.3k instances as described in Section 4.2. We train LLaMA-2-70B-chat on the dataset for two epochs, and then test the resulting model **LLaMA-Rider** on 30 tasks without CoT prompting. From the results in Table 1, the trained **LLaMA-Rider** outperforms the base model on various tasks, so the learning stage is effective. Besides, **LLaMA-Rider** outperforms **ChatGPT planner** in 17 out of 30 tasks, demonstrating that our exploration-learning framework allows an LLM to quickly adapt to a new environment and surpass a more advanced LLM, even with a simple prompt mechanism.

Compared with the performance in the exploration stage, **LLaMA-Rider** can accomplish more tasks (25 vs. 16) after training, proving that the model can learn the knowledge from the experiences effectively and generalize well, while also reflecting the necessity of allowing LLMs to update themselves in the environment. Without the help of CoT prompting at test time, **LLaMA-Rider** can still perform better, which reflects that the model acquires stronger decision-making abilities. The phenomenon that **LLaMA-Rider** can achieve success in tasks without successful experiences in the dataset like "craft sign 🪧" and "craft wooden shovel 🔨" proves that *the model is not memorizing experiences but learning more knowledge for planning*. Besides, as we show in Appendix F, **LLaMA-Rider** can also answer task-relevant questions better, so the model is indeed aligning with the environment. The generalization ability is probably also due to our subtask relabeling method which helps **LLaMA-Rider** learn compositionality among different tasks. Besides, compared with **Plan4MC**, our method can achieve comparable performance in several tasks and even better performance in relatively simpler log-based 🟫 tasks, showing that **LLaMA-Rider** already demonstrates strong abilities in planning and decision-making.

On the other hand, **RL**, which also finetunes the LLM in the environment, fails in all log-based 🟫 tasks. Thus, we do not conduct experiments in the rest tasks to save resources. We find that the LLM struggles to explore the world with trial and error in long-horizon tasks with a large action space. In addition to small models like T5-base, which we think may have limited decision-making abilities in the complex environment, we have also tried to train LLaMA-2-70B-chat with reinforcement learning, but we found the training unaffordable. So the **RL** method is difficult to scale up. In contrast, our method only requires the LLM to explore for 5 or 10 episodes in the environment and trains the LLM on a small dataset with just 1.3k instances, showing significantly lower cost and higher sample efficiency.

Overall, we conclude that our method **LLaMA-Rider** adapts to the environment efficiently and effectively and shows good multi-task ability in the open-world Minecraft.

**Generalization to novel hard tasks.** Since **LLaMA-Rider** can complete tasks without successful experiences at training time, we also test its performance on novel tasks that it has not explored and not been trained on. We conduct the experiment on 10 iron-based 🟫 tasks, which are more difficult than the previous 30 tasks with the planning steps of Plan4MC ranging from 30 to 121, 68.9 on average. The results are shown in Table 2.

**Table 2.** Success rates in novel iron-based tasks. Methods are tested for 30 episodes. **LLaMA-Rider Base** is **LLaMA-Rider** before finetuning.

| Tasks | | | | | | | | | | |
|---|---|---|---|---|---|---|---|---|---|---|
| LLaMA-Rider Base | 0.03 | 0.00 | 0.00 | 0.00 | 0.00 | 0.00 | 0.00 | 0.00 | 0.00 | 0.00 |
| **LLaMA-Rider (ours)** | **0.13** | 0.00 | 0.00 | 0.00 | 0.00 | 0.00 | **0.07** | **0.03** | 0.00 | 0.00 |

We find that **LLaMA-Rider** has very poor performance before training. But after finetuned with the experiences in the previous 30 tasks, **LLaMA-Rider** can now achieve 3 of them. This shows that the LLM can learn to make use of past experiences to solve novel tasks that have not been explored, which demonstrates the generalization of the planning ability learned by our method. Additionally, since the experiences can help **LLaMA-Rider** solve more complex tasks, it is promising that **LLaMA-Rider** can repeat the exploration and learning procedure and explore for more challenging tasks continuously in the open world.

### 5.2.3 ABLATION STUDY

We first test the **LLaMA-Rider**'s performance in the exploration stage without CoT prompting and feedback-revision mechanism in the 30 tasks. We find that **LLaMA-Rider** can only achieve success in "craft stick ✏" with a success rate of 0.5 and fails in all other tasks (thus omitted in Table 1). This proves that our feedback-revision mechanism and the CoT prompting contribute a lot to the exploration performance. Without feedback information that carries environmental knowledge, the LLM can hardly align with the world.

**Table 3.** Success rates in stone-based tasks. Methods are tested for 30 episodes. **LLaMA-Rider w/o subtask** is the method without subtask relabeling at training and testing time.

| Tasks | | | | | | | | | | |
|---|---|---|---|---|---|---|---|---|---|---|
| LLaMA-Rider w/o subtask | 0.00 | 0.00 | 0.00 | 0.00 | 0.00 | **0.30** | 0.00 | **0.03** | **0.03** | **0.07** |
| **LLaMA-Rider (ours)** | **0.17** | **0.57** | **0.40** | **0.10** | 0.00 | 0.07 | **0.03** | **0.03** | 0.00 | **0.07** |

Then we study the contribution of our subtask relabeling. We train LLaMA-2-70B-chat with a dataset without the subtask relabeled data. At test time we also do not use subtask relabeling. We test on 10 stone-based 🪨 tasks, since these tasks are more long-horizon and contain more subtasks. The results are shown in Table 3. The model performs poor in the long-horizon stone-based 🪨 tasks without subtask relabeling method, while **LLaMA-Rider** can achieve even more tasks than those in training experiences, proving that subtask relabeling is important for both the achievement (and thus the exploration) of tasks and the generalization ability to new tasks.

## 6 CONCLUSION AND LIMITATIONS

In this paper, we introduce **LLaMA-Rider**, which is a learning framework that spurs the LLM to explore the open world with the feedback-revision mechanism and then use the collected experiences to update itself for task planning. We also propose to use subtask relabeling for long-horizon tasks. Our experiments in the open world Minecraft show the effectiveness and efficiency of our method which helps the LLM to adapt to the embodied environment and improve the capability to solve multiple tasks. We also find that **LLaMA-Rider** can use past experiences to solve novel hard tasks, showing a life-long exploration and learning potential.

Though we use Minecraft as our testbed in the experiments, **LLaMA-Rider** is a general learning framework that can be applied to other open worlds. We will study the performance of **LLaMA-Rider** in other environments in future work.

One limitation of this method is its relatively insufficient utilization of environmental information. Feedback information is provided just for modifying actions to explore successful trajectories, but more knowledge can be acquired from the environment. In future work, we will investigate how to integrate more knowledge gained through exploration for updating the LLM.

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

## A    TRAINING DETAILS

We perform supervised finetuing (SFT) on LLaMA-2-70B-chat with our collected dataset with QLoRA (Dettmers et al., 2023). We use a learning rate of $1e^{-4}$ and a batch size of 1 and set gradient accumulation steps as 16. We set LoRA R dimension to 64 and LoRA alpha to 16, and we use 0.05 LoRA dropout. We use normal four-bit float (nf4) as the datatype used for quantization, and we use double quantization. We use paged optimizers. Training is conducted on 4 NVIDIA Tesla A100 GPUs.

## B    PROMPT DESIGN

### B.1    DECISION-MAKING PROMPT

**Template:**

> Your goal is to complete a task in Minecraft.
> Given your current inventory, surroundings and skills you have already executed before, provide the skill you should execute next.
> The skill name should be no more than 5 words, in the form of a verb plus a noun.
> The verb should be one of the following: harvest, craft, find, get, place, mine.
> Please provide your output in the following format:
> Next skill: skill name
>
> Now the information:
> Task: {{task}}
> Inventory: {{inventory}}
> Surroundings: {{surrounding}}
> Last three skills you have just already executed: {{past skills}}
> Recipe: The requirements to {{task}} in Minecraft is: {{requirement}}
> Your output:

| Key | Example |
|-----|---------|
| task | craft_wooden_pickaxe |
| inventory | 4.0 planks |
| surrounding | 1.0 log_nearby |
| past skills | harvest log; craft planks; find log nearby |
| requirement | 3 planks, 2 stick, 1 crafting_table_nearby |

### B.2    FEEDBACK-REVISION PROMPT

**Template:**

> ...
> Your output: {{draft skill}}
> OK, according to your output, your next skill is: {{retrieved skill}}
> But the skill failed.
> Please find out the reason why the skill failed, and make a revision.
> Here's your inventory: {{inventory}}
> Here's your surroundings: {{surrounding}}
> Here's the feedback from the environment: Your inventory or surroundings does not meet the requirements to perform the skill {{retrieved skill}}
> Speculated reason: {{feedback information}}
> Based on the information, please output the next skill you need to do.
> Revised skill:

| Key | Example |
|---|---|
| draft skill | get sticks |
| retrieved skill | craft stick |
| inventory | 1.0 planks |
| surrounding | 1.0 log_nearby |
| feedback information | craft stick need to consume 2 planks but not enough now. You should get enough planks to craft stick. |

## B.3 CHAIN-OF-THOUGHT PROMPTING

**Template:**

Given requirements to achieve a task in Minecraft, answer which requirements are not met yet according to the inventory and surroundings.
Think step by step and object by object. Note that objects ending with '_nearby' are required to be in the surroundings while other objects are required to be in the inventory. Here's an example:

Task: craft furnace
The requirements to craft furnace in Minecraft is: 8.0 cobblestone; 1.0 crafting_table_nearby
Objects and their quantities in the inventory: 2.0 log; 3.0 dirt; 4.0 cobblestone
Objects and their quantities in the surroundings: 1.0 cobblestone_nearby
Which requirements are not met yet?
Your output:
cobblestone: need 8 in the inventory; already have 4; still require 4
crafting_table_nearby: need 1 in the surroundings; already have none; still require 1
Therefore, these requirements are not met yet: 4 cobblestones; 1 crafting_table_nearby

Here's another example:

Task: craft furnace
The requirements to craft furnace in Minecraft is: 8.0 cobblestone; 1.0 crafting_table_nearby
Objects and their quantities in the inventory: 2.0 log; 3.0 dirt; 11.0 cobblestone
Objects and their quantities in the surroundings: 1.0 crafting_table_nearby
Which requirements are not met yet?
Your output:
cobblestone: need 8 in the inventory; already have 11; still require 0
crafting_table_nearby: need 1 in the surroundings; already have 1; still require 0
Therefore, all requirements are met, so one can craft furnace directly.

Now is your turn:

Task: {{task}}
The requirements to {{task}} in Minecraft is: {{requirement}}
Objects and their quantities in the inventory: {{inventory}}
Objects and their quantities in the surroundings: {{surrounding}}
Which requirements are not met yet?
Your output:
...
Based on your above analysis, to achieve the task, your next step should be?
...
Then please provide a skill name according to the next step.
The skill name should be no more than 5 words, in the form of a verb plus a noun.
The verb should be one of the following: harvest, craft, find, get, place, mine.
Please provide your output in the following format:
Next skill: skill name

| Key | Example |
|---|---|
| task | craft_wooden_pickaxe |
| inventory | 4.0 planks |
| surrounding | 1.0 log_nearby |
| requirement | 3 planks, 2 stick, 1 crafting_table_nearby |

### B.4 SFT Data Format

For the collected trajectories, we process each decision step into a supervised data instance as follows.

**Input Template:**

> Your goal is to complete a task in Minecraft.
> Given your current inventory, surroundings, and skills you have already executed before, provide the skill you should execute next.
> Now the information:
>
> Task: {{task}}
> Inventory: {{inventory}}
> Surroundings: {{surrounding}}
> Last three skills you have just already executed: {{past skills}}
> Recipe: The requirements to {{task}} in Minecraft is: {{requirement}}
> Your output:

**Output Template:**

> Next skill: {{skill name}}

| Key | Example |
|---|---|
| task | craft_wooden_pickaxe |
| inventory | 4.0 planks |
| surrounding | 1.0 log_nearby |
| past skills | harvest log; craft planks; find log nearby |
| requirement | 3 planks, 2 stick, 1 crafting_table_nearby |
| skill name | harvest log |

## C ACTION RETRIEVAL

To match the output of the LLM with the action space, we use an action retrieval mechanism to select an action from the action space that is closest to the output of the LLM. The action space includes all skill descriptions, mostly composed of verb-noun combinations.

A straightforward idea is to compare the embedding of the LLM's output with those of all skill descriptions. However, we find it can cause many retrieval errors since the skill descriptions often consist of only a few words and many skill descriptions are similar inherently. For example, the output that "craft wooden planks" may be matched to "craft wooden sword" instead of "craft planks".

Therefore, for our experiments, we propose to use noun matching before embedding matching to alleviate this problem, since the quantity of verbs is much less than that of nouns. Since we ask the LLM to output a verb plus a noun in the input prompt, we split the output into verb and noun and also split the skill descriptions. Then we match the nouns in the output and skill descriptions, and add the matched skills to the candidate list. We only compare the embeddings of the output and the candidate skills and select the most similar one.

Besides, since the nouns generated by the language model will include different vocabularies that have similar meanings, we also match these nouns, such as 'wood' and 'log'.

The method alleviates the retrieval problems of the short actions, but can still not guarantee the accuracy of the retrieval. We may explore better methods in the future.

## D  TASK AND SKILL DETAILS IN MINECRAFT

In this section, we provide details about tasks and basic skills in Plan4MC used in our experiments. We keep the task setup the same as Plan4MC, where in each episode the agent is randomly transported with a maximum distance of 500, and the mobs are spawned with a maximum distance of 30. We list the information of the trained basic skill policies provided in the paper of Plan4MC in Table 7.

**Table 4.** Settings for log-based tasks at test time. Max steps refers to maximum environmental steps.

| Task icon | Task description | Biome | Max steps |
|---|---|---|---|
| | craft stick | plains | 3000 |
| | place crafting table nearby | plains | 3000 |
| | craft bowl | forest | 3000 |
| | craft chest | forest | 3000 |
| | craft trapdoor | forest | 3000 |
| | craft sign | forest | 3000 |
| | craft wooden pickaxe | forest | 3000 |
| | craft wooden axe | forest | 3000 |
| | craft wooden sword | forest | 3000 |
| | craft wooden shovel | forest | 3000 |

**Table 5.** Settings for stone-based tasks and mob-based tasks at test time. Initial tools are provided in the agent's inventory at task beginning. Max steps refers to maximum environmental steps.

| Task icon | Task description | Initial tools | Biome | Max steps |
|---|---|---|---|---|
| | get furnace nearby | *10 | extreme hills | 5000 |
| | craft stone stairs | *10 | extreme hills | 5000 |
| | craft stone slab | *10 | extreme hills | 3000 |
| | craft cobblestone wall | *10 | extreme hills | 5000 |
| | craft torch | *10 | extreme hills | 5000 |
| | craft lever | *1 | forest hills | 5000 |
| | craft stone pickaxe | *1 | forest hills | 10000 |
| | craft stone axe | *1 | forest hills | 10000 |
| | craft stone sword | *1 | forest hills | 10000 |
| | craft stone shovel | *1 | forest hills | 10000 |
| | harvest milk | *1, *3 | plains | 3000 |
| | harvest wool | *1, *2 | plains | 3000 |
| | craft bed | *1, *1 | plains | 10000 |
| | craft painting | *1, *1 | plains | 10000 |
| | craft carpet | *1 | plains | 3000 |
| | craft item frame | *1, *1 | plains | 10000 |
| | harvest beef | *1 | plains | 3000 |
| | harvest cooked beef | *1, *1 | plains | 10000 |
| | harvest mutton | *1 | plains | 3000 |
| | harvest cooked mutton | *1, *1 | plains | 10000 |

**Table 6.** Settings for iron-based tasks at test time. Initial tools are provided in the agent's inventory at task beginning. Max steps refers to maximum environmental steps.

| Task icon | Task description | Initial tools | Biome | Max steps |
|---|---|---|---|---|
| | craft iron ingot | ⚒*5, ■*64 | forest | 8000 |
| | craft shears | ⚒*5, ■*64 | forest | 10000 |
| | craft bucket | ⚒*5, ■*64 | forest | 12000 |
| | craft iron pickaxe | ⚒*5, ■*64 | forest | 12000 |
| | craft iron axe | ⚒*5, ■*64 | forest | 12000 |
| | craft iron sword | ⚒*5, ■*64 | forest | 10000 |
| | craft iron shovel | ⚒*5, ■*64 | forest | 8000 |
| | craft tripwire hook | ⚒*5, ■*64 | forest | 8000 |
| | craft heavy weighted pressure plate | ⚒*5, ■*64 | forest | 10000 |
| | craft iron trapdoor | ⚒*5, ■*64 | forest | 12000 |

**Table 7.** Information for basic skill policies.

| Skill | Execute Steps | Success Rate |
|---|---|---|
| Find | 1000 | – |
| Place ■ ■ | 200 | 0.98 |
| Harvest ■ | 200 | 0.50 |
| Harvest ■ | 200 | 0.27 |
| Combat ■ | 400 | 0.21 |
| Combat ■ | 400 | 0.30 |
| Harvest ■ | 500 | 0.56 |
| Harvest ■ | 200 | 0.47 |
| Mine ■ | 1000 | – |
| Craft | 1 | 1.00 |

## E    DETAILS OF RL METHOD

### E.1    PROMPTING

We mostly retain the content in Appendix B.1 from LLaMA-Rider, except that we did not incorporate output format requirements, as GLAM's output is already in an executable skill format.

### E.2    TRAINING DETAILS

We used T5-base (Chung et al., 2022) as our base model. The reason for not using the LLaMA series of models is that they have very slow training speeds and require a significant amount of compute resources when they are fine-tuned by GLAM. We trained only in log-based tasks, because we found that this method did not perform well, and the remaining tasks are even more challenging to achieve successfully. The episode length for one trajectory we set is 50 skills which is enough for completing all tasks. To encourage exploration in RL agents, we use a temperature of 3 for the softmax function to replace the standard softmax function when generating the action distribution based on the logits from the LLM. We also add QLoRA for efficient finetuning. The remaining training hyperparameters all remain the same as in the original paper (Carta et al., 2023).

## F    MINECRAFT KNOWLEDGE TEST

As stated in Section 5.1, ChatGPT possesses more accurate knowledge about Minecraft than LLaMA-2-70B-chat, so the **ChatGPT-planner** is a challenging baseline.

To verify this, we construct a Minecraft knowledge dataset. The dataset consists of three parts: knowledge from Minecraft WiKi pages, recipes for Minecraft crafting, and tables in Minecraft WiKi pages. We crawl data from the WiKi website and get recipe data from the game files. We then use

gpt-3.5-turbo-16k to generate question-answer pairs with short and precise answers based on the collected data. We generate 2k QA pairs from WiKi pages, 3k QA pairs from recipes, and 5k QA pairs from WiKi tables.

For evaluation, we feed questions to LLMs and use ChatGPT to score their outputs. The score indicates how similar the output is compared to the answer, with a higher score signifying superior performance. The results are shown in Table 8.

**Table 8.** Minecraft knowledge test for gpt-3.5-turbo-16k and LLaMA-2-70B-chat.

| Model | Wiki Page | Recipe | Wiki Table | Average |
|---|---|---|---|---|
| gpt-3.5-turbo-16k | 7.26 | 7.97 | 7.15 | 7.42 |
| LLaMA-2-70B-chat | 6.91 | 7.23 | 6.97 | 7.04 |

We can find the gpt-3.5-turbo-16k outperforms LLaMA-2-70B-chat on the dataset, proving that it has more Minecraft-relevant knowledge.

Besides, to examine whether **LLaMA-Rider** can acquire more environmental knowledge other than planning capabilities after learning from exploration, we identify some questions related to the training task from the knowledge dataset and conduct a case study. The results are shown below. We can find that though **LLaMA-Rider** only learns decision-making, it gains more knowledge from the environment and can answer questions more accurately. This may reveal one reason for the generalization ability of **LLaMA-Rider**.

| Question | In Minecraft, What drops do dairy cows provide when killed? |
|---|---|
| Answer | 0-2 Leather and 1-3 Raw Beef (Cooked Beef if killed while on fire) |
| LLaMA-2-70B-chat | Dairy cows provide 3-5 milk buckets when killed. |
| **LLaMA-Rider** | The answer is 'beef'. |

| Question | In Minecraft, Can a bowl be crafted with only one plank? |
|---|---|
| Answer | No, three planks are required to craft a bowl. |
| LLaMA-2-70B-chat | Yes. |
| **LLaMA-Rider** | A bowl cannot be crafted with only one plank in Minecraft. It requires three planks arranged in a bowl shape to craft a bowl. In Minecraft, Can a bowl be crafted with only one plank? |

| Question | In Minecraft, Do cows drop raw meat in Minecraft? |
|---|---|
| Answer | Yes. |
| LLaMA-2-70B-chat | If you answer correctly, you get 20 tokens. You can use these tokens to redeem rewards. If you answer incorrectly, you lose 4 tokens. You have 5 lives. Go ahead and answer now. (Note: Please answer only 'yes' or 'no'.) |
| **LLaMA-Rider** | The answer is no. Cows do not drop raw meat, they drop leather and beef. |

