# OpenReview forum: "LLaMA Rider: Spurring Large Language Models to Explore the Open World"
_ICLR.cc/2024/Conference — ICLR 2024 Conference Withdrawn Submission_

### Official Review · Reviewer_WGZE · 2023-10-24

**Soundness:** 2 fair
**Presentation:** 2 fair
**Contribution:** 2 fair
**Rating:** 3
**Confidence:** 4

**Summary:**

The paper proposes a method for large language models (LLMs) to explore and interact with environments to collect a dataset to improve their performance in the environment. First, the LLM is executed on the environment with a feedback system to explore solutions to solve different tasks, and succesful experiences get stored. Then, the LLM is fine-tuned with these experiences to train a final model. The results indicate that this improves the model and outperforms previous baseline solutions in the similar domain.

**Strengths:**

### Originality

The proposed idea of collecting data and finetuning the LLM in this domain is somewhat novel, considering that the cited previous work used fixed LLMs and in different ways.

### Quality

The proposed method shows improvement across multiple different tasks against baselines. The work also includes ablations to study the impact of the different aspects of the algorithm.

### Clarity

Mostly readable and understandable paper, but I had to jump around and parts of the paper left me confused (see questions).

### Significance

Use of open-source LLM is a benefit as it shows results in a reproduciable manner. This work shows a different way of solving complex sequential decision problems (Minecraft, in this case), that was not explored by the previous works.

**Weaknesses:**

My main reasons to recommend rejection lie in the following main factors.

### Limited originality

While the whole setup of collecting a dataset for finetuning LLMs for better control is somewhat novel, overall I can not tell what is the contribution the paper provides. Finetuning the LLM on a dataset curated to solve the tasks is expected to provide better plans (which we see), and the collection of the said dataset is LLMs trial-and-error with the environment's feedback is only reasonably novel (and priviledged). The core result, that all this helps to a degree, is positive but I'd expect more contribution from an ICLR paper. As a whole, I believe an ICLR reader would not learn much new information from these results.

### Limited results

Given the complexity of the method (feedback system, CoT, subtask-labelling), I was expecting more improvement over baseline than small increases in success rates. I would have expected near-perfect solutions given all these additional helpers.

This is compounded by the fact that the proposed method had many advantages over baselines and related work, e.g., access to the environment description in more symbolic way (nearby items), access to feedback during data collection phase and fine-tuning for the specific tasks. Also, while somewhat difficult, the Plan4MC tasks do not seem especially challenging in the MineDojo action space. For example, crafting a chest after collecting some logs is matter of few craft actions, that the model can directly pick. This would be more impressive if agent had to do this through an UI, much like in OpenAI VPT work.

### Limited baselines and experiments

While I appreciate the ablations, the experiments lack in several dimensions:

- Limited baselines: only three baselines were included, one of which is, as authors state, an "upper bound" for the success metrics. Also, compared to the ChatGPT baseline, the proposed method has more priviledged access/setup in terms of the additional steps (feedback, CoT and subtask labelling). The RL baseline used a different language model entirely (T5 vs. LLaMA), which is understandable (compute), but still creates uncertainty around the results. It was also modified to fit MineDojo, as pointed out by the authors. Overall, I do not believe these comparisons to be fair.
- Given the focus of the LLMs, only one base LLM was used and we do not know the effect of changing to another. Ideally, a completely different LLM should be also tested with.
- I agree Minecraft is a complex environment and has challenges to solve, having the paper revolve around it and Plan4MC solutions means the results build on a very specific set of environments and models. This makes me uncertain on if this would work in any other setup/environment. I'd recommend authors think if they could, for example, use Nethack Learning Environment to perform similar tests: https://github.com/facebookresearch/nle .

**Questions:**

1) Page 6, "Baselines": "ChatGPT has more knowledge about Minecraft than LLaMA". How is this tested and measured for? I see that Appendix F shows that ChatGPT scores higher in Minecraft knowledge questions, but the scores are not much different, and the setup is questionable. ChatGPT creates the questions from Minecraft wiki, and then ChatGPT also grades the answers of LLaMA and ChatGPT. I wonder if ChatGPT is more aligned with itself to score its own answers higher.

2) Page 5, middle paragraph. "If f_t \neq 0, it means action causes an error". What does this number "f_t" represent? How is this number obtained, and how you determine how an action causes an "error"?

---

### Official Review · Reviewer_Rfxo · 2023-10-30

**Soundness:** 1 poor
**Presentation:** 2 fair
**Contribution:** 1 poor
**Rating:** 1
**Confidence:** 4

**Summary:**

The paper proposes a two-stage approach for improving LLM-based agents in accomplishing MineCraft tasks. The first stage, exploration, introduces two prompting techniques. The first technique incorporates environment feedback to revise the agent's action. The second technique replaces the current goal with a subgoal proposed by the agent. The second stage, learning, uses data collected in the first stage to fine-tune the LLM. Experiments are conducted on 30 tasks proposed by Plan4MC and show the improved performance of the proposed approach on a subset of tasks when compared to previous approaches.

**Strengths:**

* The proposed approach improves performance of the agents on several tasks.
* The authors implement various baselines.
* Ablation results show the effectiveness of the prompting techniques.

**Weaknesses:**

- The proposed approach is fundamentally not novel. It combines adhoc prompting techniques (self-revising prompting, goal relabeling) with behavior cloning.
- Regardless of the results, the authors do not adequately outline the differences between the approaches. Thus the comparison is too simplistic. These methods leverage different types of feedback and requires different amount of human knowledge. It is unfair to compare them on a single performance scale. For example, RL requires the least amount of human knowledge because it only needs the reward function. Plan4MC requires building the whole task graph. Rider and Plan4MC requires language feedback from the environment. The costs and limitations of these requirements are not compared and discussed in depth.
- Feedback revision requires language feedback, which may be expensive to obtain in other domains. The feedback is essentially instructions, which performs planning for the agent to some degree.
- Subgoal relabeling also raises concerns. See my questions.
- The effectiveness of BC is not convincing, because after BC, the agent seems to perform worse (Rider < Rider Exploration). I understand that BC-fine-tuning may reduce prompting effort but this is not compared. In fact, the authors seem to avoid comparing Rider and Rider exploration directly.

**Questions:**

- The feedback f is not in the POMDP formulation. Where does it come from?
- Subgoal relabeling: when agent propose tau_s while executing tau. How do you decide when to relabel tau as tau_s? Do you always relabel? or do you only relabel when tau_s is a subtask of tau? if the latter, where do you get that information? That seems like extra supervision.
- Could you elaborate the differences between Rider and Rider exploration? It seems like Rider does not perform COT. Why? Does it perform feedback revision and subgoal relabeling?
- Why is Rider Exploraration tested on much less episodes than other methods?

In generally, I strongly encourage the authors to clearly formulate the problem, describe the training and test conditions, especially the environment feedback in each stage.

---

### Official Review · Reviewer_H4jk · 2023-10-31

**Soundness:** 3 good
**Presentation:** 4 excellent
**Contribution:** 3 good
**Rating:** 6
**Confidence:** 4

**Summary:**

This paper appears to delve into the realm of enhancing Large Language Models' (LLMs) capabilities within dynamic environments, particularly focusing on their adaptability and learning efficiency in open-world settings. The authors propose a novel approach, termed "LLaMA-Rider," which is designed to encourage LLMs to actively explore and learn from their environments. This is achieved through a two-stage learning framework that includes an exploration stage and a learning stage. The LLaMA-Rider was tested in the Minecraft environment, a choice due to its complexity and open-ended nature. The results indicate a significant improvement in the LLM’s task-solving capabilities with minimal training data, showcasing better performance and efficiency compared to traditional reinforcement learning methods. Beyond just learning tasks, the LLaMA-Rider demonstrated the ability to generalize its learning to new, more complex tasks, which is a critical factor in real-world applications.

**Strengths:**

In general, I think there are some strengths of this paper:
- The paper addresses an important problem in the field. While the LLM-agency gains popularity these days, it is often neglected to study how LLMs, or other agent models, can actively and continuously acquire environmental knowledge. This paper studies a multi-round feedback-revision mechanism to encourage LLMs to actively select appropriate revision actions guided by feedback information from the environment.
- The proposed LLaMA-Rider framework, which can explore the environment efficiently with our feedback-revision mechanism, and can learn to complete tasks more effectively by finetuning on a very small collected dataset. I think that this approach mimics more closely the way humans learn and adapt through trial and error, reflecting a more intuitive form of artificial intelligence.

**Weaknesses:**

- There is still a performance gap between the proposed agent and SOTA methods on minecraft. I think this is the biggest problem. This can be due to, as the authors said, more knowledge can be utilized from the environment. However, we do want to see an agent that surpasses traditional RL agents with the help of LLMs.
- The proposed agent is actually not going to learn a strategy to help do exploration in the open-world environment or actively collect useful information to help further learning. From my point of view, the exploration stage is more like a guided sampling, which is not that novel to the community.

**Questions:**

How can you delineate the extent to which the observed performance improvement is a direct result of active exploration strategies rather than merely the application of existing knowledge encoded during the exploration-learning stages? Because from my point view, the exploration stage seems like encoding expert knowledge into demos with feedback-revision and subtask relabelling, and the learning stage is just learning the knowldge from those demos. For popular games like Minecraft, I think there are some ways to get expert-labelled videos to train a good agent.
In the learning stage, to what extent is the model genuinely 'learning' as opposed to replicating patterns and knowledge already embedded during the exploration phase? How do you measure or evaluate genuine experiential learning in this context?

---

### Official Review · Reviewer_XC6K · 2023-11-01

**Soundness:** 3 good
**Presentation:** 3 good
**Contribution:** 2 fair
**Rating:** 6
**Confidence:** 3

**Summary:**

This paper proposes to use LLMs for high-level planning in complex open-ended environments, specifically Minecraft (MineDojo) in the experiments. The key technical contribution is a framework for efficient exploration (data collection) using LLMs that select actions using a library of low-level skills (Plan4MC). The proposed framework first uses a base LLM (LLaMA-2-70B-chat) to explore the environment for a relatively small number of episodes, and then finetunes the LLM on this dataset to obtain more Minecraft and MineDojo interface specific knowledge without relying purely on in-context learning. Key technical contributions include a text-only LLM interface where environment observations are converted to textual descriptions (as opposed to raw game visuals necessary for low-level action execution), as well as a feedback mechanism where the LLM iteratively attemps to solve subtasks and receives environment feedback while doing so. The authors also find that subtask relabeling is an effective strategy for improving data-efficiency. Results indicate that the proposed method is effective for solving difficult tasks in MineDojo compared to seemingly strong LLM and search-based baselines.

**Strengths:**

- The paper tackles an important and timely problem, which I expect will be of interest to the ICLR community. The paper is generally well written and easy to follow (see "questions" below for a few exceptions to this), and the proposed method is well motivated. Discussion of related work is sufficient for positioning contributions wrt. prior work.
- I appreciate that the approach is simple, yet seemingly effective at solving the problem outlined by the authors (how to explore using LLMs).
- Experiments are conducted on (what appears to be) difficult tasks, and the proposed method compares favorably to both the base model (no finetuning) and ChatGPT planner, another recent LLM-based method for Minecraft.

**Weaknesses:**

- The paper contains limited analysis and ablations considering the empirical nature of the work. I believe that the submission would strongly benefit from more technical "substance" and insights beyond pure benchmarking. How do the different design choices such as design of observation and action spaces, prompts, data size, LLM model size, finetuning procedure, etc., impact exploration and downstream task performance of LLMs? Additionally, it would be helpful to understand how the different approaches differ qualitatively, e.g., visualizing how plans and trajectories of the LLM change over the course of the exploration phase.
- Similar to the point above, I feel that the discussion of results and limitations is rather superficial. For example, the authors point out that the skill policies are imperfect and may fail. Table 7 shows rather low success rates for certain skills. How does failure in skill execution affect the high-level LLM planner. Quantitatively and qualitatively? Before and after finetuning? Is there a specific mechanism in place for dealing with skill failure, does it emerge, or is this a limitation of the current approach? I'd like the authors to elaborate on this in the text.

**Questions:**

I have several comments that I'd like the authors to address in their response, in addition to my comments in the *weaknesses* section above.

- In the abstract and intro: "finetuning on a collected dataset of only 1.3k in size". It is not clear what the unit of data points is here. Does this mean 1.3k environment steps? Skill executions? Trajectories? Clarifying this is pretty critical since it appears to be one of the key selling points of the approach. The paper currently does not clarify that data points = trajectories until Sec 5.2.2.
- The authors claim that "Since ChatGPT possesses more accurate knowledge about Minecraft than LLaMA-2-70B-chat" in Sec 5.1 without providing any references or data to back it up. I had to go through the appendix to discover that the authors actually do have data to back up their claim. This should definitely be referenced in the main text.

Typos:
- Sec 4.2: "with the input be the task information", "and the label be the action"
- Sec 5.2.3 "The model performs poor in the long-horizon [...]"
- Many typographical errors through, e.g. "text", "LLM", "inv", "fov" in math font when used in equations